# Directed closure coefficient and its patterns

**Mingshan Jia** *, **Bogdan Gabrys, Katarzyna Musial**

School of Computer Science, University of Technology Sydney, Sydney, NSW, Australia

* mingshan.jia@student.uts.edu.au

## Abstract

The triangle structure, being a fundamental and significant element, underlies many theories and techniques in studying complex networks. The formation of triangles is typically measured by the clustering coefficient, in which the focal node is the centre-node in an open triad. In contrast, the recently proposed closure coefficient measures triangle formation from an end-node perspective and has been proven to be a useful feature in network analysis. Here, we extend it by proposing the directed closure coefficient that measures the formation of directed triangles. By distinguishing the direction of the closing edge in building triangles, we further introduce the source closure coefficient and the target closure coefficient. Then, by categorising particular types of directed triangles (e.g., head-of-path), we propose four closure patterns. Through multiple experiments on 24 directed networks from six domains, we demonstrate that at network-level, the four closure patterns are distinctive features in classifying network types, while at node-level, adding the source and target closure coefficients leads to significant improvement in link prediction task in most types of directed networks.

**Data Availability Statement:** All data is available at: https://github.com/MingshanJia/explore-local-structure.

**Funding:** This work was supported by the Australian Research Council, grant No.

## 1 Introduction

Networks, abstracting the interactions between components, are fundamental in studying complex systems in a variety of domains ranging from social and economic networks to informational and technological networks [1, 2]. Small subgraph patterns (also known as motifs [3] or graphlets [4]) that recur at a higher frequency than those in random networks are crucial in understanding and analysing networks. Motifs underlie many descriptive and predictive applications such as community detection [5–7], anomaly detection [8], role analysis [9], and link prediction [10, 11].

Among them, 3-node connected subgraphs, which are the building blocks for higher-order motifs, are explored most often. Further, the triangle structure, or the triadic closure [12] from a temporal perspective, has been revealed to be a natural phenomenon of networks across different areas [3, 13]. Nodes sharing a common neighbour are more likely to connect with each other. For example, in an undirected friendship network, there is an increased likelihood for two people having a common friend to become friends [14]; in a directed citation network, a paper cites two papers where one tends to cite the other [15]; and in a signed directed trust network, when Alice distrusts Bob, Alice discounts anything recommended by Bob [16].

DP190101087: "Dynamics and Control of Complex Social Networks".

**Competing interests:** The authors have declared that no competing interests exist.

The classic measure of triangle formation is the *local clustering coefficient* [17], which is defined by the percentage of the number of triangles formed with a node (referred to as node *i*) to the number of triangles that *i* could possibly form with its neighbours. Note that in this definition, the focal node *i* serves as the centre-node in an open triad [18]. To emphasise, an open triad is an unordered pair of edges sharing one node. With a focus on node *i*, it describes the extent to which edges congregate around it. As a standard metric to describe networks, the clustering coefficient has been widely used in network analysis, such as the study of community structure [19, 20], the discovery of structural role [21] and the detection of anomalous objects [22]. The extensions of local clustering coefficient have been thoroughly discussed for weighted networks [23–25], directed networks [26] and signed networks [27, 28]. Another metric for triangle formation, with a focus on an edge (referred to as $e_{ij}$ connecting node *i* and *j*), is the *edge clustering coefficient* [29] which evaluates to what extent nodes cluster around this edge.

A recent study has proposed another interesting triangle formation measure, i.e., the *local closure coefficient* [30]. With the focal node *i* as the end-node of an open triad, it is quantified as the percentage of two times the number of triangles containing *i* to the number of open triads with *i* as the end-node. Conceptually, the local clustering coefficient measures the phenomenon that two friends of mine are also friends themselves, while the local closure coefficient is focusing on a friend of my friend is also a friend of mine. This new metric has been proven to be a useful tool in several network analysis tasks such as community detection and link prediction [30]. Together with the two measures mentioned above, we propose a classification diagram of all three triangle formation measures (Fig 1). The closure coefficient is originally defined for undirected binary networks. However, in real-world complex networks, the relationships between components can be nonreciprocal (a follower is often not followed back by the followee), heterogeneous (trade volumes between countries vary significantly), and negative (an individual can be disliked or distrusted).

In this paper, from the end-node perspective, we propose the directed closure coefficient [31] to measure triangle formation in binary directed networks, and we extend it for weighted directed networks and weighted signed directed networks. Since each of the three edges can

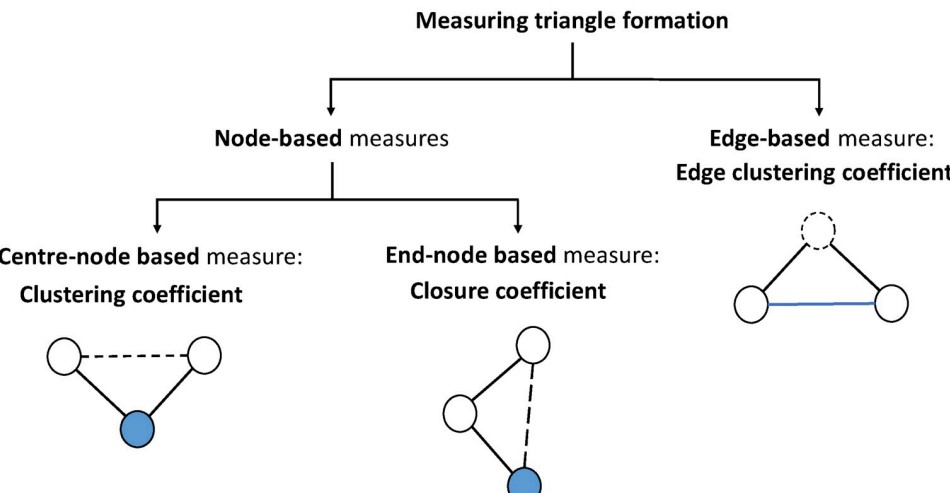

**Fig 1. Classification diagram of triangle formation measures.** In each of the two node-based measures, the focal node is painted in blue, and the dotted edge represents the potential closing edge in an open triad. In the edge-based measure, the focal edge is in blue, and the dotted outline circle represents the potential node that forms a triangle.

take either direction in a directed triangle, there are eight different triangles in total. According to the direction of the closing edge, i.e., the edge that closes an open triad and forms a triangle, we classify them into two groups (emanating from or pointing to the focal node, as shown in Fig 2a). Based on that, we propose the source closure coefficient and the target closure coefficient, respectively.

Further, from a transitive perspective, we categorise all directed triangles into four patterns: (i) a head-of-path pattern, where the focal node is at the beginning of a length-2 path; (ii) a mid-of-path pattern, where the focal node serves as an intermediate node in a length-2 path; (iii) an end-of-path pattern, where the focal node is the endpoint of a length-2 path; (iv) a cyclic pattern, where the focal node is in a cyclical path (Fig 2b). The definitions of the four closure patterns are given accordingly. Comparably, the classic directed clustering coefficient can also be categorised into four patterns [26, 32], which are found to be useful features in classifying directed networks.

Our evaluations have revealed some interesting properties of the proposed directed closure coefficient and its patterns. Through a correlation analysis on various networks, it is shown that the directed closure coefficient provides different information than the classic directed clustering coefficient on measuring triangle formation. Besides, the correlations among the eight patterns (four closure patterns and four clustering patterns) show that many types of directed networks demonstrate distinctive characteristics.

We further apply the proposed coefficients into two machine learning tasks. First, at network-level, it is shown that adding the four closure patterns in network classification improves the accuracy significantly. Also, through an analysis of feature importance, we show that compared to the four clustering patterns, the four closure patterns are more important features in classifying networks. Second, in a link prediction task, we show that at node-level, the source and target coefficients can be used together with common neighbours as effective predictors to improve the performance, especially in food webs, software networks, web graphs and word adjacency networks.

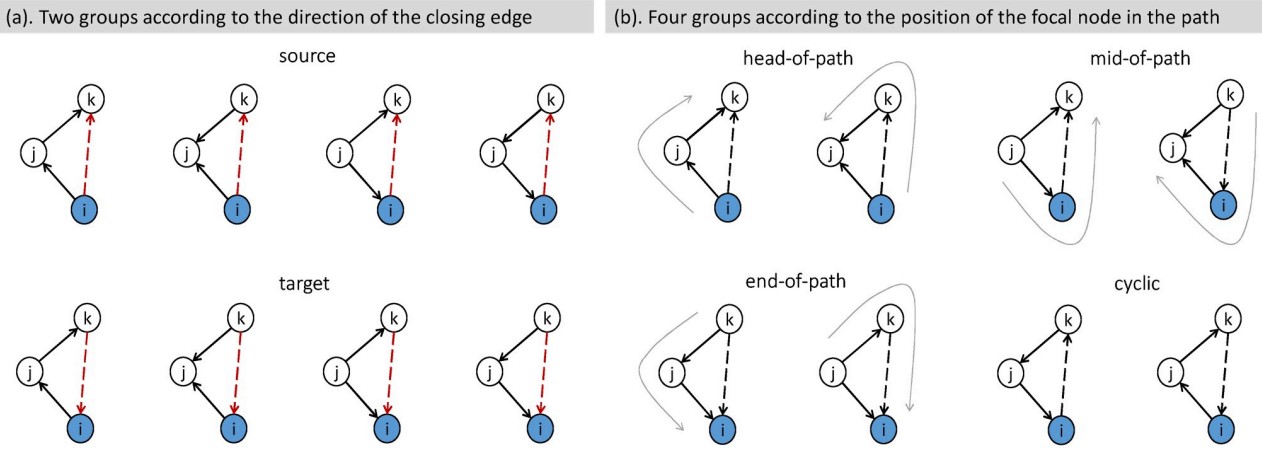

**Fig 2. Taxonomy of directed triangles.** Two solid edges connecting nodes *i, j* and *k* form an open triad, which is closed by a dotted edge connecting nodes *i* and *k*. Focal node *i*, painted in blue, is the end-node of an open triad. (a) Eight triangles are classified into two groups according to the direction of the closing edge. First row shows a group where the focal node serves as the source node of the closing edge; second row shows another group where the focal node serves as the target. (b) Eight Triangles are classified into four groups from a transitive perspective. In six transitive triads, three different patterns are distinguished by the position of node *i* in a length-2 path (emphasised by grey curved arrows): head-of-path, mid-of-path, and end-of-path patterns. The remaining two non-transitive triads are classified as a cyclic pattern.

In summary, we propose 1) the directed closure coefficient as another measure of triangle formation in directed networks (and the extension of it into weighted and signed networks); 2) the source closure coefficient and the target closure coefficient; and 3) the four closure patterns from a transitive perspective. Through multiple experiments, we exhibit the intrinsic properties of the proposed metrics and how they can be used to improve some common network analysis tasks.

## 2 Preliminaries

This section introduces the preliminary knowledge of our work, including the classic clustering coefficient, its extension in directed networks, and the closure coefficient.

### 2.1 Clustering coefficient

The clustering coefficient was originally proposed to measure the cliquishness of a neighbourhood in an undirected graph [17].

Let $G = (V, E)$ be an undirected graph on a node set $V$ (the number of nodes is $|V|$) and an edge set $E$, without multiple edges and self-loops. The adjacency matrix of $G$ is denoted as $\mathbf{A} = \{a_{ij}\}$. $a_{ij} = 1$ if there is an edge between node $i$ and node $j$, otherwise $a_{ij} = 0$. We denote the degree of node $i$ as $d_i = \sum_j a_{ij}$.

For any node $i \in V$, the *local clustering coefficient* is calculated as the number of triangles formed with node $i$ and its neighbours (labelled as $T(i)$), divided by the number of open triads with $i$ as the centre-node (labelled as $OTC(i)$):

$$C(i) = \frac{T(i)}{OTC(i)} = \frac{\frac{1}{2}\sum_j\sum_k a_{ij}a_{ik}a_{jk}}{\frac{1}{2}d_i(d_i - 1)}. \tag{1}$$

We assume that $C(i)$ is well defined. Clearly, $C(i) \in [0, 1]$.

In order to measure clustering or triangle formation at the network-level, the *average clustering coefficient* is introduced by averaging the local clustering coefficient over all nodes (an undefined local clustering coefficient is treated as zero): $\overline{C} = \frac{1}{|V|}\sum_{i \in V} C(i)$.

Another option to measure triangle formation at the network-level is the *global clustering coefficient* [33], which is defined as the fraction of open triads that form triangles in the entire network:

$$C = \frac{\sum_i\sum_j\sum_k a_{ij}a_{ik}a_{jk}}{\sum_{i \in V} d_i(d_i - 1)}. \tag{2}$$

Note that the global clustering coefficient is not equivalent to the average clustering coefficient. On some occasions, they can be very distinct from each other.

### 2.2 Directed clustering coefficient

Fagiolo [26] proposed an extension of the local clustering coefficient to directed networks, which considers all possible directed triangles formed around a focal node. In total, there are eight different triangles (each of the three edges can have two directions). When a directed open triad (or a directed triangle) contains bidirectional edges, they are treated as a combination of open triads (or triangles) with only unidirectional edges (Fig 3).

Let us denote $\mathbf{A} = \{a_{ij}\}$ as the adjacency matrix of a directed graph $G^{\mathcal{D}} = (V, E)$. $a_{ij} = 1$ if there is an edge from node $i$ to node $j$, otherwise $a_{ij} = 0$. The degree of node $i$ is denoted as $d_i$,

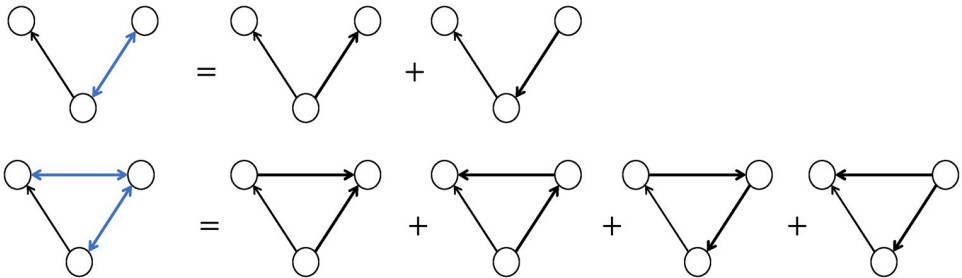

**Fig 3. Dealing with bidirectional edges.** First row shows that an open triad with one bidirectional edge is counted as two unidirectional open triads; second row shows that a triangle with two bidirectional edges is counted as four unidirectional triangles.

including both outgoing edges and incoming edges: $d_i = d_i^{out} + d_i^{in} = \sum_j a_{ij} + \sum_j a_{ji}$. $d_i^{\leftrightarrow}$ denotes the degree of bidirectional edges of $i$: $d_i^{\leftrightarrow} = \sum_j a_{ij} a_{ji}$.

The *local directed clustering coefficient* is thus defined as the number of directed triangles formed with node $i$ and its neighbours (counted as unidirectional ones, labelled as $T^{\mathcal{D}}(i)$), divided by twice the number of directed open triads with $i$ as the centre-node (labelled as $OTC^{\mathcal{D}}(i)$):

$$C^{\mathcal{D}}(i) = \frac{T^{\mathcal{D}}(i)}{2OTC^{\mathcal{D}}(i)} = \frac{(1/2)\sum_j\sum_k (a_{ij} + a_{ji})(a_{ik} + a_{ki})(a_{jk} + a_{kj})}{d_i(d_i - 1) - 2d_i^{\leftrightarrow}}. \tag{3}$$

Note that $OTC^{\mathcal{D}}(i)$ equals to $(1/2)[d_i(d_i - 1) - 2d_i^{\leftrightarrow}]$. $OTC^{\mathcal{D}}(i)$ is multiplied by two because the edge closes a directed open triad can take two directions.

Similarly, the *average directed clustering coefficient* of the entire network is defined as: $\overline{C^{\mathcal{D}}} = |V|^{-1}\sum_{i \in V} C^{\mathcal{D}}(i)$. An expected alternative, i.e., the global directed clustering coefficient has not appeared in literature. We therefore give a definition here:

**Definition 2.1**. The ***global directed clustering coefficient*** of a directed network, denoted $C^{\mathcal{D}}$, is defined as the fraction of directed open triads that form triangles in the entire network:

$$C^{\mathcal{D}} = \frac{\frac{1}{2}\sum_i\sum_j\sum_k \left(a_{ij} + a_{ji}\right)\left(a_{ik} + a_{ki}\right)\left(a_{jk} + a_{kj}\right)}{\sum_{i \in V}(d_i(d_i - 1) - 2d_i^{\leftrightarrow})}. \tag{4}$$

The numerator here equals three times the number of directed triangles in the entire network (each node of a triangle contributes an open triad with it as the centre-node).

## 2.3 Closure coefficient

Recently Yin et al. [30] proposed the *local closure coefficient* and thus closed a gap in measuring triangle formation on undirected networks. Different from the ordinary centre-node focus in the local clustering coefficient, this definition is based on the end-node of an open triad. Recall that an open triad is an unordered pair of edges sharing one node. For example, in an open triad $ijk$ with two edges $ij$ and $jk$, there is no difference between $(ij, jk)$ and $(jk, ij)$.

Reusing the above notations for undirected graph, the local closure coefficient of node $i$ is defined as two times the number of triangles formed with $i$ (labelled as $T(i)$), divided by the

number of open triads with $i$ as the end-node. (labelled as $OTE(i)$):

$$E(i) = \frac{2T(i)}{OTE(i)} = \frac{\sum_j\sum_k a_{ij}a_{ik}a_{jk}}{\sum_{j\in N(i)}(d_j - 1)}, \tag{5}$$

where $N(i)$ denotes the set of neighbours of node $i$. $T(i)$ is multiplied by two for the reason that each triangle contains two open triads with $i$ as the end-node. When a triangle is actually formed (e.g., with nodes $i$, $j$ and $k$), the focal node $i$ can be viewed as the centre-node in one open triad ($jik$) or as the end-node in two open triads ($ijk$ and $ikj$). Obviously, $E(i) \in [0, 1]$.

At the network-level, the *average closure coefficient* is then defined as the mean of the local closure coefficient over all nodes: $\overline{E} = \frac{1}{|V|}\sum_{i\in V}E(i)$. Analogous to the global clustering coefficient (Eq 2), we can give a global version of the closure coefficient:

$$E = \frac{2\sum_{i\in V}T(i)}{\sum_{i\in V}\sum_{j\in N(i)}(d_j - 1)}. \tag{6}$$

The numerator is equal to six times the number of triangles in the entire network (each node of a triangle contributes two open triads with it as the end-node), divided by twice the number of open triads constructed from the end-node in the entire network. However, this definition is actually equivalent to the global clustering coefficient (Eq 2) as globally the difference of the position of the focal node will not surface.

**Proposition 1**. *In any undirected network, E = C.*

*Proof.* Since globally the neighbourhood relationship is reciprocal, $\sum_{i\in V}\sum_{j\in N(i)}(d_j - 1)$ can be written as $\sum_{j\in V}\sum_{i\in N(j)}(d_j - 1)$ which equals $\sum_{j\in V}d_j(d_j - 1)$. Then we have $\sum_{i\in V}\sum_{j\in N(i)}(d_j - 1) = \sum_{i\in V}d_i(d_i - 1)$. Thus, $E = C$.

## 3 Closure coefficient in directed networks

The local closure coefficient has been proven to be a useful metric in undirected networks [30]. In this section, we first provide a general extension of it to directed networks, i.e., the local directed closure coefficient. We further propose the closure coefficients of particular patterns. Finally, we extend it into weighted (signed) directed networks.

### 3.1 Closure coefficient in binary directed networks

Motivated by the closure coefficient and the directed clustering coefficient, we aim to measure the directed triangle formation from the end-node of an open triad. There are eight different directed triangles, and similarly a triangle (or an open triad) with bidirectional edges is treated as a combination of triangles (or open triads) with only unidirectional edges (Fig 3).

Reusing the notations in Section 2, we now give the definition of the closure coefficient in directed networks.

**Definition 3.1**. The *local directed closure coefficient* of node $i$ in a directed network, denoted $E^{\mathcal{D}}(i)$, is defined as twice the number of directed triangles formed with node $i$ (labelled as $T^{\mathcal{D}}(i)$), divided by twice the number of directed open triads with $i$ as the end-node (labelled as $OTE^{\mathcal{D}}(i)$):

$$E^{\mathcal{D}}(i) = \frac{2T^{\mathcal{D}}(i)}{2OTE^{\mathcal{D}}(i)} = \frac{\sum_j\sum_k(a_{ij} + a_{ji})(a_{ik} + a_{ki})(a_{jk} + a_{kj})}{2\sum_{j\in N(i)}(a_{ij} + a_{ji})(d_j - (a_{ij} + a_{ji}))}. \tag{7}$$

When the neighbours of $i$ are solely connected to $i$, the local directed closure coefficient is undefined. In real-world networks, however, nodes with undefined closure coefficients are very rare.

$T^{\mathcal{D}}(i)$ is multiplied by two since each triangle contains two open triads with $i$ as the end-node. $OTE^{\mathcal{D}}(i)$ is multiplied by two because the closing edge of a directed open triad can take two directions. Obviously, $E^{\mathcal{D}}(i) \in [0, 1]$. When the adjacency matrix $\mathbf{A}$ is symmetric (the network becomes undirected), Eq 7 reduces to Eq 5, i.e., $E^{\mathcal{D}}(i) = E(i)$.

Similarly, in order to measure at the network-level, we propose the definition of an average directed closure coefficient and a global directed closure coefficient.

**Definition 3.2**. The ***average directed closure coefficient*** of a directed network, denoted $\overline{E^{\mathcal{D}}}$, is defined as the average of the local directed closure coefficient over all nodes:

$$\overline{E^{\mathcal{D}}} = \frac{1}{|V|}\sum_{i \in V} E^{\mathcal{D}}(i), \tag{8}$$

in which an undefined local directed closure coefficient is treated as zero. In the case of a random network, where each directed edge occurs with a probability $p$, one has that $\mathbb{E}[E^{\mathcal{D}}(i)] = p$.

**Definition 3.3**. The ***global directed closure coefficient*** of a directed network, denoted $E^{\mathcal{D}}$, is defined as:

$$E^{\mathcal{D}} = \frac{2\sum_{i \in V} T^{\mathcal{D}}(i)}{2\sum_{i \in V}\sum_{j \in N(i)}(a_{ij} + a_{ji})(d_j - (a_{ij} + a_{ji}))}, \tag{9}$$

where the numerator equals six times the number of directed triangles in the entire network (each node of a triangle contributes two open triads with it as the end-node), divided by twice the number of directed open triads across the network.

Similar to Proposition 1 and its proof, the global directed closure coefficient is equivalent to the global directed clustering coefficient (Eq 4).

**Proposition 2**. *In any directed network*, $E^{\mathcal{D}} = C^{\mathcal{D}}$.

## 3.2 Closure coefficients of particular patterns

In addition to a general measure, we propose to have a closer look at the directed closure coefficients of particular patterns in order to gain a deeper understanding and fully realise the potential of this metric.

Recall that in Fig 2(a), when a directed triangle is constructed from an end-node-based open triad, the closing edge is incident to the focal node. Therefore, we propose to classify directed triangles into two groups according to the direction of the closing edge: one group where the focal node serves as the source node of the closing edge, another group where the focal node serves as the target. Two definitions are given accordingly.

**Definition 3.4**. For a given node $i$ in a directed network, the ***source closure coefficient***, denoted $E^{src}(i)$, and the ***target closure coefficient***, denoted $E^{tgt}(i)$ are defined as:

$$E^{src}(i) = \frac{T^{src}(i)}{OTE^{\mathcal{D}}(i)} = \frac{\sum_j\sum_k(a_{ij} + a_{ji})(a_{jk} + a_{kj})a_{ik}}{\sum_{j \in N(i)}(a_{ij} + a_{ji})(d_i - (a_{ij} + a_{ji}))}, \tag{10}$$

$$E^{tgt}(i) = \frac{T^{tgt}(i)}{OTE^{\mathcal{D}}(i)} = \frac{\sum_j\sum_k(a_{ij} + a_{ji})(a_{jk} + a_{kj})a_{ki}}{\sum_{j \in N(i)}(a_{ij} + a_{ji})(d_i - (a_{ij} + a_{ji}))}. \tag{11}$$

$T^{src}(i)$ indicates the number of triangles where the focal node acts as the source node of the closing edge. Since we view triangles as being built from an end-node perspective, certain

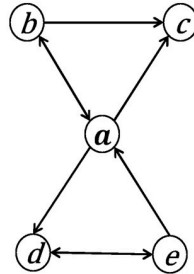

| | $T^D$ | $T^{src}$ | $T^{tgt}$ | $OTE^D$ | $E^D$ | $E^{src}$ | $E^{tgt}$ |
|---|---|---|---|---|---|---|---|
| $a$ | 4 | 5 | 3 | 7 | 0.57 | *0.71* | *0.43* |
| $b$ | 2 | 3 | 1 | 7 | 0.29 | *0.43* | *0.14* |
| $c$ | 2 | 0 | 4 | 6 | 0.33 | *0* | *0.67* |
| $d$ | 2 | 1 | 3 | 6 | 0.33 | *0.17* | *0.5* |
| $e$ | 2 | 3 | 1 | 6 | 0.33 | *0.5* | *0.17* |

**Fig 4. An example of calculating the source closure coefficient and target closure coefficient.**

triangles (the ones where the focal node has two outgoing edges) are counted twice. Thus, $0 \leq T^{src}(i) \leq 2T^D(i)$. Similarly, $T^{tgt}(i)$ denotes the number of triangles where the focal node acts as the target node.

Comparing these two equations with the directed closure coefficient(Eq 7), the denominators are $OTE^D(i)$ instead of $2OTE^D(i)$. This is because the closing edge here can only take one direction, either outgoing or incoming, thus ensuring the two definitions are in the range of [0, 1]. It is obvious that $T^{src}(i)+T^{tgt}(i) = 2T^D(i)$, which then gives $E^{src}(i) + E^{tgt}(i) = 2E^D(i)$. On a small network, Fig 4 shows how the source closure coefficient and the target closure coefficient are calculated in a detailed table.

These two metrics evaluate the extent to which the focal node acts as the source node or the target node of the closing edges in a triangle formation. Note that there are no analogous definitions for the clustering coefficient because the closing edge is not incident to the focal node that serves as the centre-node of the open triad. In Section 4.3, we show how the source and target closure coefficients can be used to improve the performance in a link prediction task.

Furthermore, several studies have shown that the three-node transitive closure (also called the feedforward loop) prevails in many real-world networks [3, 13, 26, 32]. Thus, we propose to categorize the eight directed triangles into four patterns from a transitive perspective: three transitive patterns distinguished by the position of the focal node in a length-2 path, plus one non-transitive pattern (Fig 2b). Before introducing the definitions of directed closure coefficients of these four patterns, we first characterize four types of directed open triads with the focal node as the end-node; a comparison with centre-node focused triads is also provided (Fig 5). Then we give the following definitions.

**Definition 3.5**. The directed closure coefficients of four patterns, i.e., the **head closure coefficient**, denoted $E^{head}(i)$; the **end closure coefficient**, denoted $E^{end}(i)$; the **mid closure**

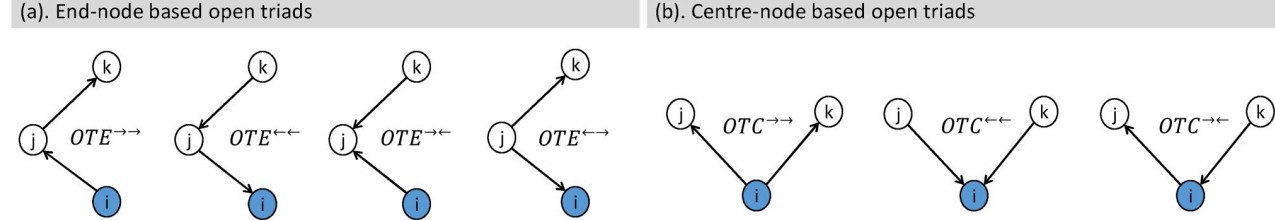

**Fig 5. Two groups of directed open triads.** (a). Four different open triads with the focal node $i$ as the end-node. Two arrows on the superscript describe the directions of two edges: First arrow depicts an edge from $i$ to $j$ ($\rightarrow$) or from $j$ to $i$ ($\leftarrow$); second arrow depicts an edge from $j$ to $k$ ($\rightarrow$) or from $k$ to $j$ ($\leftarrow$). (b). Three different open triads with $i$ as the centre-node. First arrow depicts the edge direction between $i$ and $j$ while second arrow depicts the edge direction between $i$ and $k$. There are three instead of four since when the focal node is in the centre, node $j$ and $k$ are symmetric to it.

*coefficient*, denoted $E^{mid}(i)$ and the *cyclic closure coefficient*, denoted $E^{cyc}(i)$ are defined as:

$$E^{head}(i) = \frac{2T^{head}(i)}{OTE^{\rightarrow\rightarrow}(i) + OTE^{\rightarrow\leftarrow}(i)} = \frac{\sum_j\sum_k a_{ij}a_{ik}(a_{jk} + a_{kj})}{\sum_{j\in N(i)} a_{ij}(d_j - (a_{ij} + a_{ji}))},$$

$$E^{end}(i) = \frac{2T^{end}(i)}{OTE^{\leftarrow\leftarrow}(i) + OTE^{\leftarrow\rightarrow}(i)} = \frac{\sum_j\sum_k a_{ji}a_{ki}(a_{jk} + a_{kj})}{\sum_{j\in N(i)} a_{ji}(d_j - (a_{ij} + a_{ji}))},$$

$$E^{mid}(i) = \frac{2T^{mid}(i)}{OTE^{\rightarrow\leftarrow}(i) + OTE^{\leftarrow\rightarrow}(i)} = \frac{\sum_j\sum_k (a_{ji}a_{ik}a_{jk} + a_{ki}a_{ij}a_{kj})}{\sum_{j\in N(i)} (a_{ij}(d_j^{in} - a_{ij}) + a_{ji}(d_j^{out} - a_{ji}))},$$

$$E^{cyc}(i) = \frac{2T^{cyc}(i)}{OTE^{\rightarrow\rightarrow}(i) + OTE^{\leftarrow\leftarrow}(i)} = \frac{\sum_j\sum_k (a_{ji}a_{ik}a_{kj} + a_{ki}a_{ij}a_{jk})}{\sum_{j\in N(i)} (a_{ji}(d_j^{in} - a_{ij}) + a_{ij}(d_j^{out} - a_{ji}))}.$$

As shown above, the numerator of each coefficient equals twice the number of particular triangles; the denominator can be calculated with the neighbourhood information of node $i$ and the degree information of $i$'s neighbours. Fagiolo [26] and Ahnert [32] also proposed four patterns for the directed clustering coefficient. In order to better compare the four closure patterns with the four clustering patterns, we briefly list their equations here:

$$C^{head}(i) = \frac{T^{head}(i)}{2OTC^{\rightarrow\rightarrow}(i)}, \quad C^{end}(i) = \frac{T^{end}(i)}{2OTC^{\leftarrow\leftarrow}(i)},$$

$$C^{mid}(i) = \frac{T^{mid}(i)}{OTC^{\rightarrow\leftarrow}(i)}, \quad C^{cyc}(i) = \frac{T^{cyc}(i)}{OTC^{\rightarrow\leftarrow}(i)}.$$

The significance of defining the four closure patterns is twofold. First, at node-level analysis, they can be applied directly to measure whether a node of interest is more of an initiator (higher value of the head closure coefficient), an intermediary (higher value of the mid closure coefficient) or a target (higher value of the end coefficient). Secondly, after averaging over all nodes, the four closure patterns can also serve as features at network-level. Section 4.2 shows how the four closure patterns are used in a supervised learning task to classify different types of directed networks.

## 3.3 Closure coefficient in weighted networks

So far, the study is focusing on binary networks, where the value of every edge is either one or zero. In many networks, however, we need a more accurate representation of the relationships between nodes, such as the frequency of contact in a social network, the traffic flow in a road network, etc. This is why we are also interested in defining a closure coefficient for weighted networks.

We begin with weighted undirected networks. Several versions of weighted clustering coefficients have been summarised in [34]. Among them, a definition given by Onnela et al. [24] and another given by Zhang and Horvath [25] are often employed. After normalisation (maximum weight normalised to one), the former takes a geometric average of weights of actually formed triangles, divided by the number of potential triangles, which implies all edges taking the maximum weight in the denominator. The latter chooses a simple product of weights of formed triangles, divided by the product of two weights of an open triad, implying the potential triadic closing edge taking the maximum weight.

In our definition of weighted closure coefficient, similar to the method proposed by Zhang and Horvath [25], we choose to assign a maximum weight to the closing edge. In a weighted graph $G^{\mathcal{W}}$ described by its weight matrix $\mathbf{W} = \{w_{ij}\}$, we suppose $w_{ij} \in [0, 1]$ (normalised by the maximum weight), and the strength of node $i$ is $s_i = \sum_j w_{ij}$.

**Definition 3.6**. The ***weighted closure coefficient*** of node $i$ in a weighted network, denoted $E^{\mathcal{W}}(i)$, is defined as:

$$E^{\mathcal{W}}(i) = \frac{\sum_j \sum_k w_{ij} w_{ik} w_{jk}}{\sum_{j \in N(i)} w_{ij}(s_j - w_{ij})}. \tag{12}$$

Obviously, $E^{\mathcal{W}}(i) \in [0, 1]$. When the weight matrix becomes binary, Eq 12 degrades to Eq 5, i.e., $E^{\mathcal{W}}(i) = E(i)$.

In a similar approach, the definition of closure coefficient in weighted directed networks can be extended from Eq 7. Let us denote $\mathbf{W} = \{w_{ij}\}$ as the weight matrix of a weighted directed graph $G^{\mathcal{W},\mathcal{D}}$, $w_{ij} \in [0, 1]$. The strength of node $i$ is denoted by $s_i$ ($s_i = \sum_j w_{ij} + \sum_j w_{ji}$). Then we have the following definition:

**Definition 3.7**. The ***weighted directed closure coefficient*** of node $i$, denoted $E^{\mathcal{W},\mathcal{D}}(i)$, is defined as:

$$E^{\mathcal{W},\mathcal{D}}(i) = \frac{\sum_j \sum_k (w_{ij} + w_{ji})(w_{ik} + w_{ki})(w_{jk} + w_{kj})}{2 \sum_{j \in N(i)} (w_{ij} + w_{ji})(s_j - (w_{ij} + w_{ji}))}. \tag{13}$$

This definition can also be used in weighted signed networks ($w_{ij} \in [-1, 1]$), with a modified definition of $s_i$ ($s_i = \sum_j |w_{ij}| + \sum_j |w_{ji}|$). In many settings, the weights of relationships can be both positive and negative, as a person may trust or distrust others with different levels of intensity. Clearly, $E^{\mathcal{W},\mathcal{D}}(i)$ varies in $[-1, 1]$. It is positive when positive triangles formed around the focal node outweigh negative ones. It equals zero when no triangles are formed with the focal node or positive triangles and negative triangles are balanced.

Through a brief case study on the Bitcoin Alpha trust network (TR-BTCAlpha) [35], we illustrate how the weighted directed closure coefficient can provide new understandings in network analysis. TR-BTCAlpha is a trust network on a blockchain asset trading platform, where users rate other traders in a range of $[-10, 10]$ in steps of 1, from total distrust to total trust. There are 3,783 nodes representing the users and 24,186 edges representing the ratings in the network.

First, without considering weights on edges, we find in Fig 6a that the directed closure coefficient is positively related to the node degree (Pearson correlation coefficient $\rho$ equals to 0.714), implying big traders tend to form more trustful cliques. However, when weights are put back, we see in Fig 6b that the correlation between the weighted directed closure coefficient and the node strength becomes very weak ($\rho = 0.265$): big traders are not significantly better at forming trustful cliques. Also, we detect some nodes with negative closure coefficients, meaning the negative triangles outweigh the positive ones around them. In line with the balance theory [36] suggesting that negative triangles are rare in a trust relationship, we find only 138 out of 3783 nodes having formed overall distrustful cliques.

## 3.4 Computational efficiency

To end this section, we briefly discuss the computational efficiency of the proposed metrics. Taking the local directed closure coefficient (Definition 3.1) as an example, for the purpose of facilitating understanding and expression, we use the adjacency matrix of the network to present the equation, which leads up to $O(|V|^2)$ in computation.

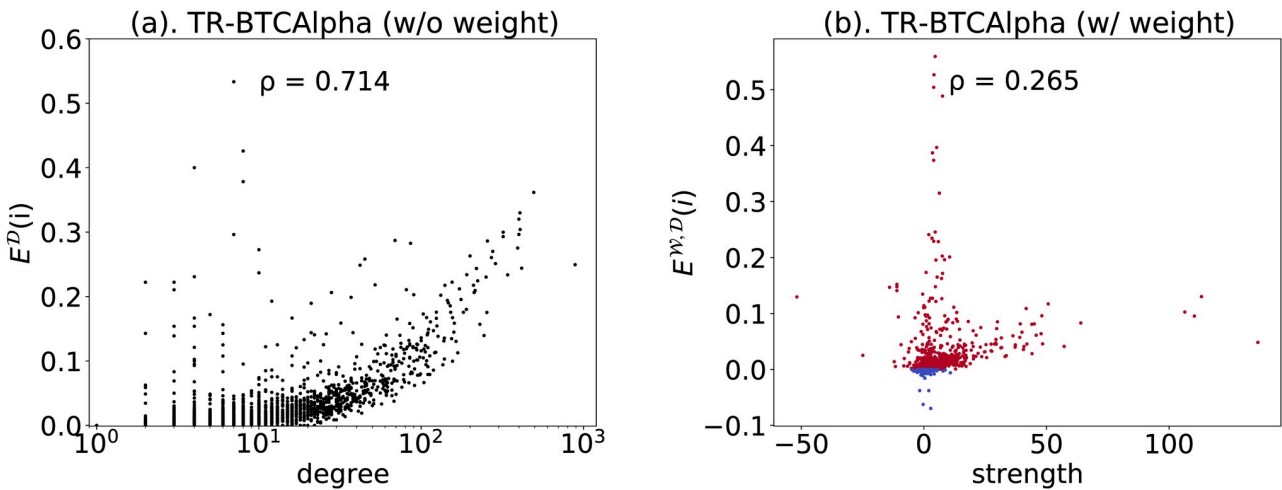

**Fig 6. Case study of the network TR-BTCALPHA.** (a). The correlation between directed closure coefficient and node degree (weights ignored). All nodes are plotted in black dots; (b). The correlation between weighted directed closure coefficient and node strength (weights taken into account). 3654 nodes with positive closure coefficients are plotted in red; 138 nodes with negative closure coefficients are plotted in blue.

In actual development, however, after conveniently obtaining the neighbourhood information (both successors and predecessors in directed networks) of a given node, the average-case computational cost is $O(\overline{k}^2)$, where $\overline{k}$ is the average degree of the network. The average cost for computing the local directed closure coefficient across the network is therefore $O(|V| \cdot \overline{k}^2)$. As in most real networks $\overline{k} \ll |V|$, the computation is fast in large networks.

## 4 Experiments and analysis

In this section, we evaluate the proposed directed closure coefficient and its patterns in real-world networks. First, we compare it with the classic directed clustering coefficient. Then, we demonstrate that at network-level, the four closure patterns are discriminative in classifying directed networks. Finally, at node-level, we show how the source and target closure coefficients can be applied in link prediction task. Our code is available at https://github.com/MingshanJia/explore-local-structure.

### 4.1 Directed closure coefficient in real-world networks

**4.1.1 Datasets.** We run experiments on 24 directed networks from 6 different domains:

1. Four trust networks. TR-BTCALPHA and TR-BTC-OTC [37]: two who-trusts-whom networks of users on Bitcoin trading platforms Bitcoin Alpha and Bitcoin OTC; TR-ADVOGATO [38]: a network of trust relationships among users on an online community Advogato; TR-EPINIONS [39]: a who-trust-whom network of members on a general consumer review site Epinions.com.

2. Four food webs. FW-MANGROVE [40]: a what-eats-what network among species found in Florida's mangroves during the wet season; FW-BAYWET and FW-BAYDRY [41]: two food webs collected from the cypress wetlands of South Florida during the wet season and the dry season; FW-LITTLEROCK [42]: a food web among the species found in Little Rock Lake in Wisconsin.

3. Four citation networks. CITCORA [43]: citations among papers indexed by CORA; CITHEPPH [44] and CITHEPPH [45]: citations among papers posted on arxiv.org under the hep-ph and hep-th categories, between 1993 and 2003; CITCITESSER [46]: citations among papers indexed by the CiteSeer digital library.

4. Four software networks. SW-WEKA [47]: a class dependency network of the Weka 3.6.6 framework; SW-LUCENE [48]: a class dependency network of the Lucene 4.1.0 framework; SW-JUNG [47]: a class dependency network within the JUNG 2.0.1 and javax 1.6.0.7 library namespaces; SW-JDK [47]: a class dependency network within the JDK 1.6.0.7 framework.

5. Four web graphs. WEB-STANFORD [49]: a hyperlink network of Stanford University; WEB-NOTREDAME [50]: a hyper link network of the University of Notre Dame; WEB-BERKSTAN [49]: a hyperlink network between UC Berkeley and Stanford University; WEB-GOOGLE [49]: a hyperlink network of a portion of the general WWW released in 2002 by Google.

6. Four word adjacency networks. WA-JAPANESE, WA-ENGLISH, WA-FRENCH and WA-SPANISH [51]: directed networks of word adjacency in texts of languages including Japanese, English, French and Spanish.

Table 1 lists some key statistics of these datasets. We see that in all 24 networks, the average directed closure coefficient is smaller than the average directed clustering coefficient. That is to say, in all these types of networks, fewer triangles are built from end-node-based open triads than from centre-node-based open triads. In food webs, the difference between them is not very big; while in word adjacency networks and two software networks (SW-JUNG and JDK), the directed closure coefficient is several dozen times smaller than the directed clustering coefficient.

From the scatter plots of the local directed closure coefficient and the local directed clustering coefficient (Fig 7), we can see their relationship more clearly. First, in most networks covered in our study, the two coefficients have a positive Pearson correlation whereas only in five networks they show negative correlation. However, neither positive nor negative correlations are strong, ranging from −0.296 to 0.518, indicating the directed closure coefficient captures different information on triangle formation than the classic directed clustering coefficient. Secondly, the same type of a network exhibits a similar relationship between the two variables. A visual inspection of Fig 7 indicates that the plots within one type of network (in the same row) are more similar to each other than the plots in between these types of networks (between rows). For example, in citation networks, most points are congregated at the left bottom area; and in word adjacency networks, most nodes have relatively small directed closure coefficient, making most points very close to the horizontal axis.

We further explore the correlations amongst the eight patterns, i.e., the four closure patterns and the four clustering patterns. In Fig 8, we observe that certain types of networks demonstrate particular characteristics. Specifically, in trust networks, we find strong correlations among almost all four closure patterns (except between $E^{head}$ and $E^{end}$). Also, $C^{mid}$ and $C^{cyc}$ are strongly correlated. In citation networks, two mid-of-path patterns ($C^{mid}$ and $E^{mid}$) and two cyclic patterns ($C^{cyc}$ and $E^{cyc}$) are highly correlated. In software networks, two end-of-path patterns ($C^{end}$ and $E^{end}$) and two cyclic patterns ($C^{cyc}$ and $E^{cyc}$) have higher correlations. In web graphs, the correlation between $E^{head}$ and $E^{mid}$ and the correlation between $E^{end}$ and $E^{cyc}$ are stronger. At last, in word adjacency networks, the four closure patterns are strongly correlated with each other. These observations lead us to the following experiment, in which we utilise these patterns as features to classify different types of directed networks.

**Table 1. Statistics of datasets, showing the number of nodes ($|V|$), the number of edges ($|E|$), the average degree($\bar{k}$), the proportion of reciprocal edges ($r$), the average directed clustering coefficient ($\overline{C^D}$), and the average directed closure coefficient ($\overline{E^D}$).** Datasets having timestamps on edge creation are superscripted by ($\tau$).

| Network | $|V|$ | $|E|$ | $\bar{k}$ | $r$ | $\overline{C^D}$ | $\overline{E^D}$ |
|---|---|---|---|---|---|---|
| TR-BTCAlpha$^\tau$ | 3,783 | 24,186 | 6.39 | 0.832 | 0.158 | 0.017 |
| TR-BTC-OTC$^\tau$ | 5,881 | 35,592 | 6.05 | 0.792 | 0.151 | 0.013 |
| TR-Advogato | 6,539 | 51,127 | 7.82 | 0.307 | 0.148 | 0.026 |
| TR-Epinions | 75,879 | 509K | 6.71 | 0.405 | 0.110 | 0.016 |
| FW-Mangrove | 97 | 1,492 | 15.38 | 0.062 | 0.261 | 0.185 |
| FW-BayWet | 128 | 2,106 | 16.45 | 0.029 | 0.177 | 0.134 |
| FW-BayDry | 128 | 2,137 | 16.70 | 0.029 | 0.176 | 0.135 |
| FW-LittleRock | 183 | 2,494 | 13.63 | 0.034 | 0.173 | 0.112 |
| Citcora | 23,166 | 91,500 | 3.95 | 0.051 | 0.146 | 0.055 |
| CIT-HepTh | 27,770 | 353K | 12.70 | 0.003 | 0.157 | 0.061 |
| CitHepPh | 34,546 | 422K | 12.20 | 0.003 | 0.143 | 0.053 |
| CitCitesser | 384K | 1,751K | 4.56 | 0.010 | 0.092 | 0.028 |
| SW-Weka | 1,323 | 4,844 | 3.66 | 0.014 | 0.201 | 0.021 |
| SW-Lucene | 2,956 | 10,872 | 3.68 | 0.005 | 0.217 | 0.029 |
| SW-JUNG | 6,120 | 50,535 | 8.26 | 0.010 | 0.454 | 0.006 |
| SW-JDK | 6,434 | 53,892 | 8.38 | 0.009 | 0.443 | 0.006 |
| WEB-Stanford | 282K | 2,312K | 8.20 | 0.277 | 0.378 | 0.055 |
| WEB-NotreDame | 326K | 1,497K | 4.60 | 0.507 | 0.159 | 0.029 |
| WEB-BerkStan | 685K | 7,601K | 11.09 | 0.250 | 0.400 | 0.055 |
| WEB-Google | 876K | 5,105K | 5.83 | 0.307 | 0.370 | 0.097 |
| WA-Japanese | 2,704 | 8,300 | 3.07 | 0.073 | 0.139 | 0.004 |
| WA-English | 7,381 | 46,281 | 6.27 | 0.090 | 0.252 | 0.005 |
| WA-French | 8,325 | 24,295 | 2.92 | 0.037 | 0.114 | 0.002 |
| WA-Spanish | 11,586 | 45,129 | 3.90 | 0.091 | 0.249 | 0.002 |

## 4.2 Network classification

This section presents the utility of the proposed four closure patterns in classifying different types of directed networks. Previous works have shown that a normalised number of directed triads and triangles, such as the triad significance profile [51] and the clustering signatures [32], are effective attributes in a network classification task. It motivated us to use the four closure patterns in the network classification, as they represent a normalised number of directed triangles from the end-node perspective.

In order to gain an intuitive understanding of the effect of the four closure patterns on detecting different types of networks, we draw a parallel coordinates plot (Fig 9). Without complicated conditional rules, it is clearly seen that some types of networks show discriminative profiles in terms of certain closure patterns. For example, food webs are better separated from other types of networks with respect to their head closure coefficients, word adjacent networks with respect to their end or mid closure coefficients, and web graphs or trust networks in respect of their cyclic closure coefficients.

**4.2.1 Setup.** To prepare the classification dataset, we calculate the average four clustering patterns and the average four closure patterns of each network. We then choose Decision Tree (DT) as the classifier, not only because it is a powerful algorithm, but also because it enables convenient calculation of feature importance. We also include two tree-based ensemble models that are more stable and more powerful than a single DT, i.e., the Random Forest (RF) classifier and the Gradient Boosted Decision Tree (GBDT) classifier in the experiment.

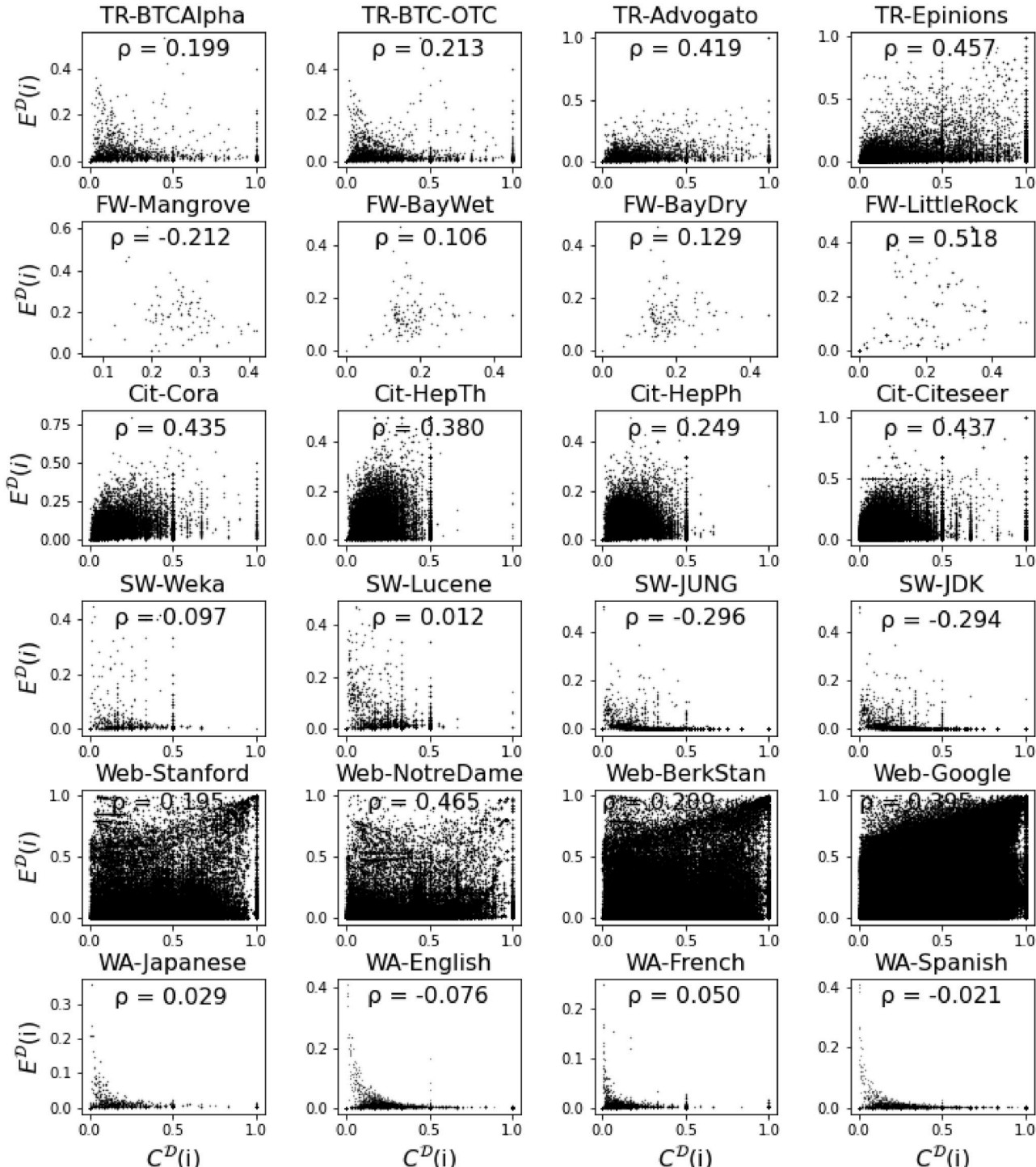

**Fig 7. Correlation between the directed clustering coefficient and the directed closure coefficient, together with the Pearson correlation coefficient $\rho$.** Each dot in the plot represents a node in the network.

In order to test how useful the four closure patterns are in classifying networks, we fit three sets of features into these models: first set, the baseline, includes the traditional four clustering patterns [26, 32]; second set includes the proposed four closure patterns; third set includes both the clustering patterns and the closure patterns. As the dataset is small, we adopt the

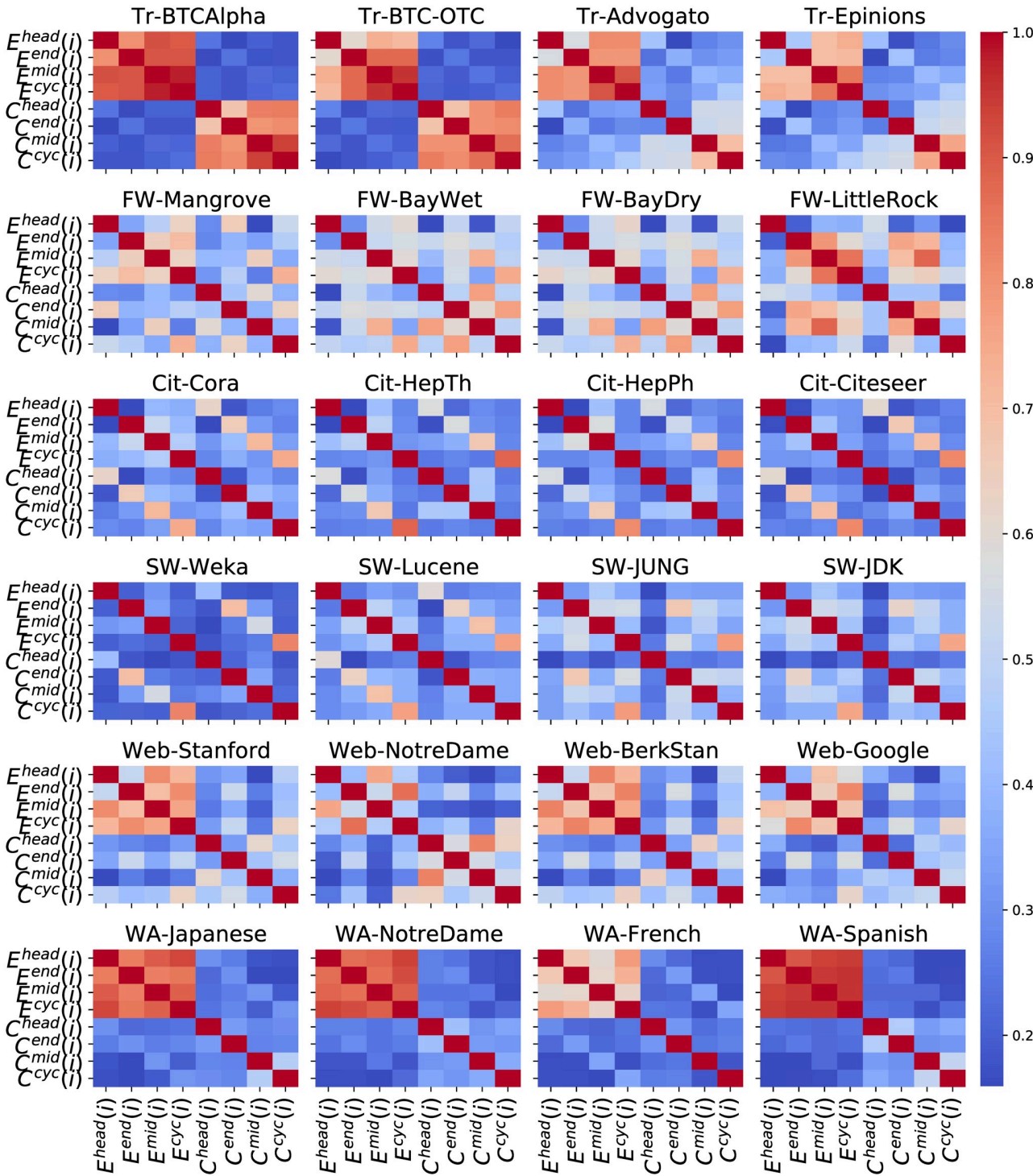

**Fig 8. Heatmap of the correlations among the eight patterns in 24 networks.**

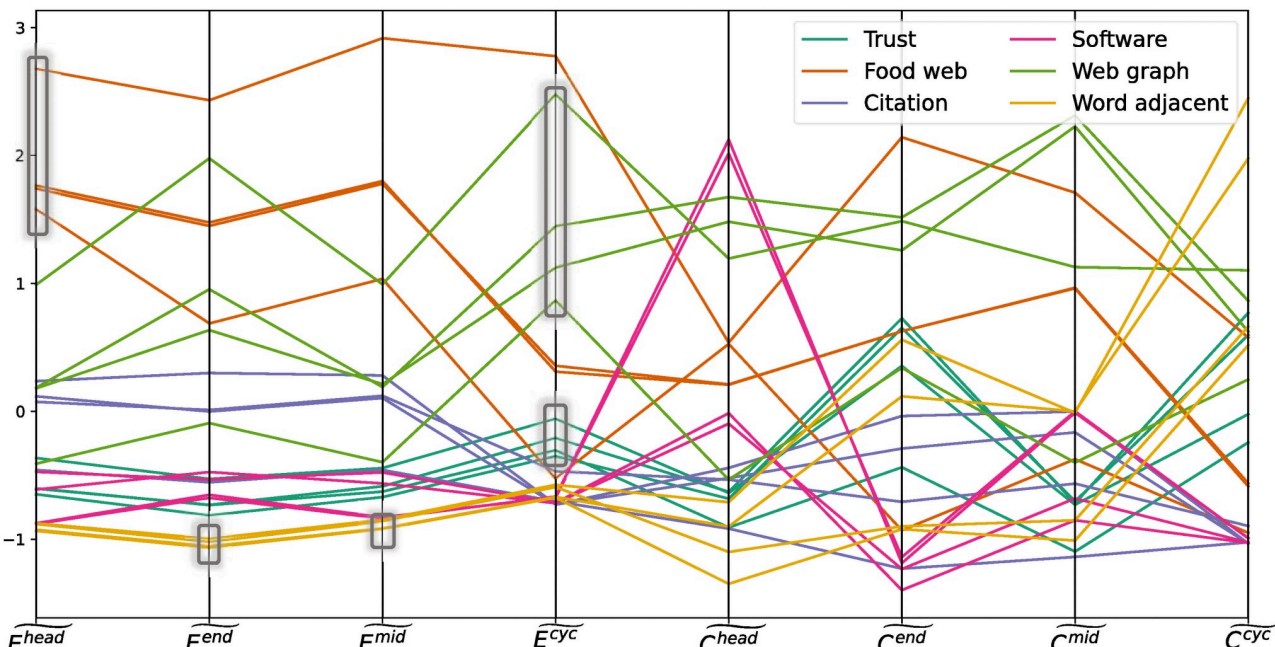

**Fig 9. Parallel coordinates plot of 24 networks on eight features, including the four closure patterns and the four clustering patterns.** Each vertical axis represents one feature. In order to put all features on a similar scale, the value of each feature is standardised by removing the mean and scaling to unit variance. Different types of networks are painted in different colours, as shown in the legend. Distinct braids of line segments are highlighted by thin rectangles.

leave-one-out cross-validation [52] to evaluate the classification performance with different set of features. Also, because tree-based models are naturally stochastic, we repeat 1000 times and report the mean accuracy score.

**4.2.2 Results and discussion.** Table 2 shows the mean accuracy of three classifiers with different sets of features. Comparing the first row and the second row, we have two classifiers (DT and GBDT) performing better with the closure patterns and one classifier (RF) performing better with the clustering patterns. In order to further study how different features influence the classification results, we take Random Forest classifier as an example and report the two average confusion matrices of using two different feature sets (Fig 10). We can see that the clustering patterns are better at classifying software networks and web graphs, while the closure patterns are better at categorising food webs and word adjacency networks. It indicates that the information contained in the closure patterns are complementary to the information contained in the clustering patterns. Therefore, combining both would be expected to yield the best classification accuracy as also illustrated in the third row of Table 2.

Indeed, comparing the first row and the third row, after adding the four closure patterns to the four clustering patterns, we observe significant improvement in all three classifiers,

**Table 2. Leave-one-out cross-validation accuracy in classifying network types.** Three sets of network features (rows) are tested in three tree-based classifiers (columns).

|  | DT | RF | GBDT |
|---|---|---|---|
| with four clustering patterns | 0.557 | 0.734 | 0.667 |
| with four closure patterns | 0.631 | 0.716 | 0.708 |
| with four clustering patterns & four closure patterns | 0.672 | 0.765 | 0.797 |

(a) Avg. error matrix of RF with clustering patterns

| Act.\Pred. | TR | FW | Cit | SW | Web | WA |
|---|---|---|---|---|---|---|
| TR | **2.96** | 0 | 0 | 0 | 0 | 1.04 |
| FW | 0 | **2** | 0 | 1 | 1 | 0 |
| Cit | 0.01 | 0.02 | **2.98** | 0.99 | 0 | 0 |
| SW | 0 | 0 | 0 | **4** | 0 | 0 |
| Web | 1 | 0 | 0 | 0 | **3** | 0 |
| WA | 1.34 | 0 | 0 | 0 | 0 | **2.66** |

(b) Avg. error matrix of RF with closure patterns

| Act.\Pred. | TR | FW | Cit | SW | Web | WA |
|---|---|---|---|---|---|---|
| TR | **3** | 0 | 1 | 0 | 0 | 0 |
| FW | 0 | **2.81** | 0.96 | 0 | 0.23 | 0 |
| Cit | 0 | 0 | **2.99** | 1 | 0.01 | 0 |
| SW | 0 | 0 | 1 | **2.38** | 0 | 0.62 |
| Web | 1 | 1 | 0 | 0 | **2** | 0 |
| WA | 0 | 0 | 0 | 0.06 | 0 | **3.94** |

**Fig 10. Average confusion matrices of Random Forest model with different feature sets.**

especially in Decision Tree and Gradient Boosted Decision Tree classifiers (more than 19%). The result demonstrates that the proposed four closure patterns are useful features in telling apart different types of directed networks.

Naturally, the next question is how important each feature is in classifying these networks. Adopting the common approach to measure feature importance in tree-based models [52], we calculate the importance score by computing the normalised total decrease in impurity brought by each feature. After repeating 1000 times, we report the average importance scores of the eight features in Fig 11. We observe that in DT and RF classifiers, all four closure patterns have larger importance scores than the four clustering patterns, and the most important feature is $E^{mid}$. In GBDT, although $C^{end}$ has the second largest importance score, overall speaking, the total score of the closure patterns is still larger than that of the clustering patterns. This analysis illustrates further that the proposed four closure patterns are important features in network classification.

## 4.3 Link prediction in directed networks

Many studies [53–58] have shown that future interactions among nodes can be extracted from the network topology information. The key idea is to compare the proximity or similarity between pairs of nodes, either from the neighbourhoods [54, 55], the local structures [56] or the whole network [57, 58].

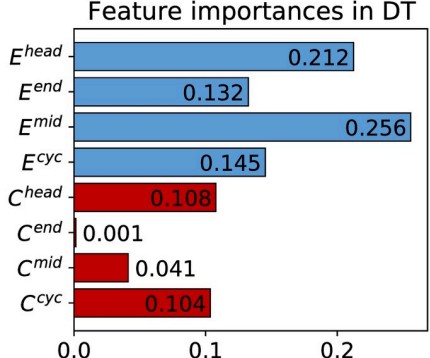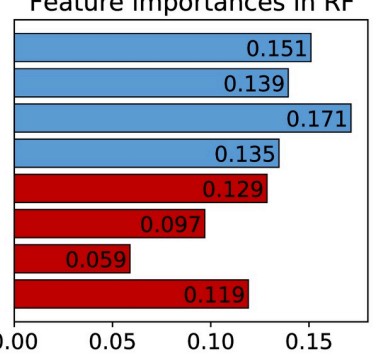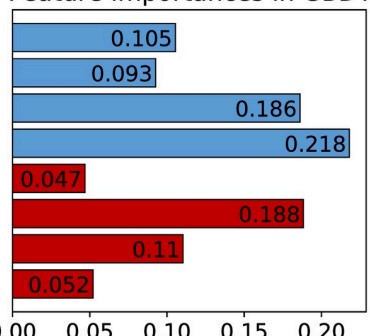

**Fig 11. Importance scores of eight features in three tree-based models classifying network types.** The scores of the four closure patterns are plotted in blue bars while those of the four clustering patterns are plotted in red bars.

Most existing methods, however, focus solely on undirected networks. In this experiment, we show whether the information provided by the local directed closure coefficient can be used to enhance the performance of link prediction approaches for directed networks. As shown in [53], the neighbourhood based methods are simple yet powerful. We choose three classic similarity indices extended for directed networks as the baseline methods [59].

Let $N_{out}(i)$ be the out-neighbour set of node $i$ (consisting of $i$'s successors); $N_{in}(i)$ be the in-neighbour set (consisting of $i$'s predecessors). The set of all neighbours $N(i)$ is the union of the two: $N(i) = N_{out}(i) \cup N_{in}(i)$. For an ordered pair of nodes $(s, t)$, the three baseline indices are defined below:

1. Directed Common Neighbours index: $DiCN(s, t) = |N_{out}(s) \cap N_{in}(t)|$,

2. Directed Adamic-Adar index: $DiAA(s, t) = \sum_{u \in N_{out}(s) \cap N_{in}(t)} \frac{1}{\log|N(u)|}$,

3. Directed Resource Allocation index: $DiRA(s, t) = \sum_{u \in N_{out}(s) \cap N_{in}(t)} \frac{1}{|N(u)|}$.

**4.3.1 Proposed indices.** Combining the idea of the Common Neighbours index and the source and target closure coefficients (Definition 3.4), we propose two indices to measure the *directed closeness* in directed networks.

**Definition 4.1**. For an ordered pair of nodes $(s, t)$, the ***closure closeness index***, denoted $CCI(s, t)$; and the ***extra closure closeness index***, denoted $ECCI(s, t)$ are defined as:

$$CCI(s, t) = |N_{out}(s) \cap N_{in}(t)| \cdot (E^{src}(s) + E^{tgt}(t)),$$

$$ECCI(s, t) = |N(s) \cap N(t)| \cdot (E^{src}(s) + E^{tgt}(t)).$$

Unlike the closure closeness index, the extra closure closeness index uses the set of all neighbours, because the source closure coefficient of node $s$ and the target closure coefficient of node $t$ can also bring in the direction inclination.

**4.3.2 Setup.** We model a directed network as a graph $G^{\mathcal{D}} = (V, E)$. For networks having timestamps on edges, we order the edges according to their appearing times and select the first 50% edges and related nodes to form an "old graph", denoted $G_{old} = (V^*, E_{old})$. For networks not having timestamps, we randomly choose 50% edges and related nodes as $G_{old}$ and repeat 10 times in the experiment. Apparently, the total number of potential links on node set $V^*$ equals to $|V^*|^2 - |E_{old}|$. Let $E_{new}$ be the set of future edges among the nodes in $V^*$. We apply each prediction method to output a list containing the similarity scores for all potential links. Each potential link either represents a positive link or a negative link, depending on whether it appears in $E_{new}$. The PR-AUC value of the prediction is then calculated. Since in very large networks it is too expensive to compute all potential links, we randomly sample 3,000 connected nodes on $G^{\mathcal{D}}$ when $|V| > 10,000$ and repeat the above procedures 10 times.

**4.3.3 Results and discussion.** We compare three baseline methods with two proposed methods (Definition 4.1) in Table 3. We see that the closure closeness index (CCI) has recorded the highest PR-AUC value in 12 networks, including all the food webs, all the software networks, three web graphs and one citation network. The extra closure closeness index (ECCI), on the other hand, has recorded the highest PR-AUC value in 6 networks, including all the word adjacency networks and two trust networks. It suggests that in most directed networks, including the local structure information of the source and target closure coefficients leads to improvement in link prediction. The improvement is remarkable in many networks: CCI is over 25% better than the baselines in three software networks (SW-WEKA, SW-JUNG and SW-JDK), and over 10% better in all four food webs and two web graphs (WEB-STANFORD

**Table 3. Performance comparison of five methods on link prediction in directed networks (PR-AUC).** The best performance in each network is in bold type, second best in italic.

| Network | DiCN | DiAA | DiRA | CCI | ECCI |
|---|---|---|---|---|---|
| TR-BTCAlpha[τ] | 0.0286 | *0.0291* | 0.0199 | 0.0283 | **0.0347** |
| TR-BTC-OTC[τ] | 0.0275 | *0.0308* | 0.0245 | 0.0265 | **0.0316** |
| TR-Advogato | 0.1076 | **0.1124** | 0.0899 | 0.1052 | *0.1107* |
| TR-Epinions | *0.1536* | **0.1559** | 0.1303 | 0.1520 | 0.1491 |
| FW-Mangrove | 0.2334 | 0.2438 | 0.2456 | **0.2760** | *0.2666* |
| FW-BayWet | 0.1669 | 0.1705 | 0.1719 | **0.1995** | *0.1903* |
| FW-BayDry | 0.1738 | 0.1771 | 0.1783 | **0.2058** | *0.1887* |
| FW-LittleRock | *0.2593* | 0.2520 | 0.2443 | **0.3117** | 0.2449 |
| CIT-Cora | **0.1084** | *0.1056* | 0.1007 | 0.1053 | 0.0819 |
| CIT-HepTh | 0.1742 | **0.1897** | 0.1769 | *0.1833* | 0.1708 |
| CIT-HepPh | *0.1428* | **0.1459** | 0.1324 | 0.1424 | 0.1339 |
| CIT-Citeseer | 0.1054 | *0.1063* | 0.1029 | **0.1221** | 0.0791 |
| SW-Weka | 0.1231 | 0.1394 | *0.1399* | **0.1901** | 0.0935 |
| SW-Lucene | *0.1853* | 0.1730 | 0.1678 | **0.1930** | 0.1026 |
| SW-JUNG | *0.3386* | 0.3277 | 0.2732 | **0.4385** | 0.1645 |
| SW-JDK | *0.3610* | 0.3377 | 0.2785 | **0.4551** | 0.1787 |
| WEB-Stanford | 0.3784 | *0.3875* | 0.3330 | **0.4159** | 0.2927 |
| WEB-NotreDame | 0.2226 | 0.2310 | 0.2104 | **0.2934** | *0.2703* |
| WEB-BerkStan | 0.4002 | *0.4026* | 0.3746 | **0.4784** | 0.3968 |
| WEB-Google | 0.4938 | **0.5211** | 0.4803 | *0.5046* | 0.3903 |
| WA-Japanese | 0.0240 | 0.0197 | 0.0154 | *0.0353* | **0.0568** |
| WA-Darwin | 0.0421 | 0.0451 | 0.0337 | *0.0654* | **0.0901** |
| WA-French | 0.0136 | 0.0152 | 0.0138 | *0.0262* | **0.0488** |
| WA-Spanish | 0.0571 | 0.0631 | 0.0537 | *0.1048* | **0.1368** |

and WEB-NotreDame). Besides, ECCI is over 100% better than the three baselines in word adjacency networks.

We also notice that in all four software networks and one citation network (CitCitesser), where CCI records the highest precision, ECCI is, however, worse than the baseline methods. This suggests that sometimes the information provided by the extra neighbours without considering direction inclination conflicts with that provided by the source and target closure coefficients. Finding a method that better combines the information of common neighbours and closure coefficients is an interesting avenue for future study.

## 5 Related work

In this section, we summarise some additional related work that also measure directed triangle formation from an end-node perspective. Similar to our work, Yin et al. [60] extended the local closure coefficient in directed networks by proposing a family of eight coefficients. Their definition of the local directed closure coefficients of node i are eight scalars $H_{xy}^z(i)$ with $x, y, z \in \{i, o\}$ ($i$ and $o$ represent edge direction, incoming or outgoing). One major limitation of this work is that it lacks a general characterisation that unifies all eight directed triangles constructed from end-node based open triads. Our work not only addressed this issue by giving one general definition, but further proposed the taxonomies of end-node based directed triangles, i.e., the source and target closure coefficients and the four closure patterns.

Romero and Klenberg [61] developed a methodology for studying a particular type of directed closure process (or "link copying" phenomenon) in information network. Lou et al. [62] later proposed a graphical model TriFG to predict reciprocity and triadic closure in social networks. Nevertheless, these two works chose not to take into account all directed triangles by particularly focusing on the feed-forward triangle.

## 6 Conclusion

In this paper, we introduced the directed closure coefficient and its patterns to measure directed triangle formation from an end-node perspective. Through experiments on 24 real-world networks from six domains, we revealed that 1) in all networks, the average directed closure coefficient is smaller than the average directed clustering coefficient; 2) the correlation between the directed closure coefficient and the directed clustering coefficient is weak; 3) different types of networks demonstrate different characteristics in the correlations of the eight patterns.

We also showed that, at network-level, adding the four closure patterns leads to significant improvement in classifying directed networks; while at node-level analysis, such as in link prediction, the source and target coefficients can be used together with common neighbours as effective predictors, especially in food webs, software networks, web graphs and word adjacency networks. Due to the simplicity and interpretability in the definitions, we anticipate that the directed closure coefficient and its patterns will become standard descriptive features and be incorporated in other network mining tasks.

## Acknowledgments

The authors thank the editors and anonymous reviewers for their excellent comments and suggestions. The authors would also thank Pim van der Hoorn, Xiaolin Zhang, Mohamad Barbar, Joakim Skarding and Yu-Xuan Qiu for their helpful comments and discussions.

## Author Contributions

**Conceptualization:** Mingshan Jia.

**Formal analysis:** Mingshan Jia.

**Funding acquisition:** Bogdan Gabrys, Katarzyna Musial.

**Investigation:** Mingshan Jia.

**Methodology:** Bogdan Gabrys, Katarzyna Musial.

**Project administration:** Katarzyna Musial.

**Software:** Mingshan Jia.

**Supervision:** Bogdan Gabrys, Katarzyna Musial.

**Writing – original draft:** Mingshan Jia.

**Writing – review & editing:** Mingshan Jia, Bogdan Gabrys, Katarzyna Musial.

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
