## [Decision Letter · Decision Letter 0]

4 May 2021

PONE-D-21-06621

Directed closure coefficient and its patterns

PLOS ONE

Dear Dr. Jia,

Thank you for submitting your manuscript to PLOS ONE. After careful consideration, we feel that it has merit but does not fully meet PLOS ONE’s publication criteria as it currently stands. Therefore, we invite you to submit a revised version of the manuscript that addresses the points raised during the review process.

Please follow the reviewers advices to revise the manuscript. 

We look forward to receiving your revised manuscript.

Kind regards,

Hocine Cherifi

Academic Editor

PLOS ONE

Journal Requirements:

Reviewers' comments:

Reviewer's Responses to Questions

**Comments to the Author**

1. Is the manuscript technically sound, and do the data support the conclusions?

Reviewer #1: Yes

Reviewer #2: Yes

2. Has the statistical analysis been performed appropriately and rigorously? 

Reviewer #1: Yes

Reviewer #2: N/A

3. Have the authors made all data underlying the findings in their manuscript fully available?

Reviewer #1: Yes

Reviewer #2: Yes

4. Is the manuscript presented in an intelligible fashion and written in standard English?

Reviewer #1: Yes

Reviewer #2: Yes

5. Review Comments to the Author

Reviewer #1: Please see attached report

(Copied several times to achieve ridiculus minimal character limit)

Please see attached report

Please see attached report

Please see attached report

Please see attached report

Reviewer #2: In this paper, the authors propose an extension of the local closure to directed networks. They introduce the closure coefficients of particular patterns. They also extend it to weighted directed networks.

This work is an excellent contribution to the fascinating study of triangles in complex networks. Going beyond the classical clustering coefficient measures, they investigate the closure coefficients for classifying the network structure and link prediction. Experiments are performed on 24 directed networks originating from various domains. They show the advantage of using the concept of closure coefficients compared to the classic clustering coefficients for these tasks. The subject of this paper is of prime interest to the network science community. The contribution is well presented. The results are sound. Therefore, I recommend its publication. I have nevertheless two minor comments that need to be addressed before that.

First of all, the authors can better highlight the paper's interest by broadening the literature review to show the importance of the issue. I suggest incorporating and commenting on literature dealing with the clustering coefficient related to the community structure.

Q. Ji, D. Li, and Z. Jin, "Divisive Algorithm Based on Node Clustering Coefficient for Community Detection," in IEEE Access, vol. 8, pp. 142337-142347, 2020, doi: 10.1109/ACCESS.2020.3013241.

Orman K., Labatut V., Cherifi H. (2013) An Empirical Study of the Relation between Community Structure and Transitivity. In: Menezes R., Evsukoff A., González M. (eds) Complex Networks. Studies in Computational Intelligence, vol 424. Springer, Berlin, Heidelberg. https://doi.org/10.1007/978-3-642-30287-9_11

See also

Trolliet T., Cohen N., Giroire F., Hogie L., Pérennes S. (2021) Interest Clustering Coefficient: A New Metric for Directed Networks Like Twitter. In: Benito R.M., Cherifi C., Cherifi H., Moro E., Rocha L.M., Sales-Pardo M. (eds) Complex Networks & Their Applications IX. COMPLEX NETWORKS 2020 2020. Studies in Computational Intelligence, vol 944. Springer, Cham. https://doi.org/10.1007/978-3-030-65351-4_48

Second, make sure that the code and the data are available to reproduce and diffuse their work more efficiently.

6. PLOS authors have the option to publish the peer review history of their article (what does this mean?). If published, this will include your full peer review and any attached files.

Reviewer #1: **Yes: **Pim van der Hoorn

Reviewer #2: No

---

## [Author Response · Author response to Decision Letter 0]

9 Jun 2021

We have uploaded our responses as a separate file.

---

## [Editor Report · Decision Letter 1]

14 Jun 2021

Directed closure coefficient and its patterns

PONE-D-21-06621R1

Dear Dr. Jia,

We’re pleased to inform you that your manuscript has been judged scientifically suitable for publication and will be formally accepted for publication once it meets all outstanding technical requirements.

Kind regards,

Hocine Cherifi

Academic Editor

PLOS ONE
---

## [Editor Report · Acceptance letter]

16 Jun 2021

PONE-D-21-06621R1 

Directed closure coefficient and its patterns 

Dear Dr. Jia:

I'm pleased to inform you that your manuscript has been deemed suitable for publication in PLOS ONE. Congratulations! Your manuscript is now with our production department. 

Kind regards, 

on behalf of

Professor Hocine Cherifi 

Academic Editor

PLOS ONE